# Increased IGF2 and Immunosuppressive Cell Populations in Ascites of Patients with Recurrent High-Grade Serous Ovarian Cancer

**DOI:** 10.3390/biomedicines13092074

**Published:** 2025-08-26

**Authors:** Laura F. Mortan, Jacqueline A. Bohn, Doris Mangiaracina Benbrook

**Affiliations:** 1Gynecologic Oncology Section, Stephenson Cancer Center, University of Oklahoma Health Sciences, Oklahoma City, OK 73104, USA; laura-mortan@ou.edu (L.F.M.); jacqueline-bohn@ou.edu (J.A.B.); 2Pathology Department, University of Oklahoma Health Sciences, Oklahoma City, OK 73104, USA; 3Obstetrics and Gynecology Department, University of Oklahoma Health Sciences, Oklahoma City, OK 73104, USA

**Keywords:** ovarian cancer, metastases, ascites, IGF2

## Abstract

**Background/Objectives**: Ovarian cancer is frequently diagnosed at advanced stages, during which ascites serves as a microenvironment conducive to cancer recurrence. This study aimed to identify factors in ascites specimens that could be targeted for the detection and prevention of recurrence of the most common and lethal histology type, high-grade serous ovarian cancer (HGSOC). **Methods**: Ascites specimens were collected from patients with HGSOC who provided informed consent. RNA was isolated from ascites cells, sequenced, and compared between ascites cells in two initial and four recurrent HGSOC samples using DRAGENv.4.2.4 to identify differentially expressed genes. Immune cell populations were estimated from the differentially expressed genes using deconvolution analysis. ELISA and antibody isotyping were used to evaluate cell-free ascitic fluid (N = 24) and banked serum (N = 23) collected from independent groups of patients with HGSOC who provided informed consent. **Results**: Transcriptomics analysis identified reduced expression of immunoglobulin variable chains in the recurrent ascites group. The primary immunoglobulin present in all ascites specimens was IgG1, whereas IgG2 and IgG3 appeared to be present at higher levels in recurrent ascites. Deconvolution analysis estimated a more suppressive immune cell population profile in the recurrent samples. The immunosuppressor insulin-like growth factor 2 (IGF2) was the most differentially expressed gene, with higher expression in recurrent ascites than in initial ascites. Significantly higher levels of IGF2 protein were measured in recurrent cell-free ascites fluid than in initial cell-free ascites fluid. **Conclusions**: In conclusion, IGF2 and suppressive immune cell populations were identified as candidate drug targets for prevention of ovarian cancer recurrence.

## 1. Introduction

Ovarian cancer is the primary cause of mortality among gynecologic malignancies in the United States. Due to the absence of effective screening methods, approximately 50% of ovarian cancer cases are diagnosed at an advanced stage [1,2]. Diagnoses are generally in women aged 55–64 who are post-menopausal, with the median age of diagnosis at 63 years old [3]. The current treatment for advanced-stage ovarian cancer is debulking surgery, paclitaxel, and a platinum agent for six cycles, followed by maintenance therapy until recurrence [4,5,6]. The introduction of poly (ADP) ribose polymerase (PARP) inhibitors and antiangiogenic therapies has improved the management of advanced and recurrent ovarian cancer [7,8,9]. Maintenance therapy is typically administered to patients after they reach an undetectable level of disease, allowing them to remain on treatment until recurrence. Patients prescribed maintenance therapy are classified as having a high risk of recurrence. Recent advances in maintenance therapies have proven effective in prolonging disease-free survival rates. Patients generally have a period of undetectable cancer; however, most experience recurrence and eventually succumb to their disease [10]. The recurrence and metastasis of ovarian cancer primarily occur through the trans coelomic route [11,12,13].

The initial development of ovarian cancer is important for understanding the spread of the disease. High-grade serous ovarian cancer (HGSOC) is the most prevalent and aggressive histological subtype of ovarian cancer, originating from serous tubal intraepithelial carcinoma lesions harboring mutations in the tumor protein 53 gene (*TP53*) or the ovarian surface epithelia [14,15,16,17]. Once cells detach from STIC lesions, they can migrate to the ovary and begin to establish solid tumors. Cells detached from tumors of the ovary most often colonize the omentum prior to disseminating throughout the peritoneum [18].

Cancer cells can detach from the primary tumor site and move through the peritoneal cavity as individual cells and in tightly clustered aggregates known as spheroids within a fluid referred to as ascites. Ascites accumulation is documented in most, but not all, cases of advanced ovarian cancer and almost every recurrence case [19,20,21,22,23]. Ascitic fluid facilitates disease spread within the peritoneal cavity, thereby promoting further fluid accumulation [20,21,22,23,24,25]. Tumor cells adhere to and disrupt the lymphatic ducts, thereby impeding fluid drainage. The presence of ascites is hypothesized to contribute to the establishment of a liquid tumor microenvironment, which facilitates tumor progression, immune evasion, and resistance to treatment [20,24]. In ascitic environments, cells aggregate to form spheroid clusters. It is hypothesized that the peripheral cell layers are exposed to the highest levels of therapeutic agents, whereas the central cell layers encounter a diminished dosage and oxygen deprivation-induced necrosis [25,26]. This differential exposure and hypoxia may lead to adaptations in cellular mechanisms, fostering resistance to future therapies. The study of ascitic fluid has recently emerged as a critical focus in ovarian cancer research, contributing to a deeper understanding of metastatic mechanisms, the efficacy of existing treatments, and the innovation of novel targeted therapies.

The primary aim of this study was to determine whether significant differences in gene and protein expression could be identified between ascites collected from patients with initial and recurrent HGSOC. Furthermore, this study sought to assess whether these differences could be utilized for the prediction, prevention, and molecular targeting of ovarian recurrence.

## 2. Materials and Methods

### 2.1. Patient Eligibility and Consent

The specimens and data utilized in this study were obtained from three protocols approved by the Institutional Review Board (IRB) of the University of Oklahoma Health Sciences. Specimens and their related data were selected for this study based on the following eligibility criteria: females aged ≥18 years with presumed or confirmed ovarian, fallopian tube, or primary peritoneal cancer accompanied by ascites, as determined by computed tomography imaging. Different sets of specimens were used for various methods. Six ascites specimens were used for the RNA sequencing analysis. Additional ascites specimens were added to this set to equal a total of 24 specimens were used for the IGF2 protein ELISA. A 16-specimen subset of these 24 samples was used to measure IgG protein levels, and a 10-specimen subset of these 24 specimens was used for the isotyping analysis. Twenty-three serum specimens from a different set of patients with HGSOC were used for another IGF2 protein ELISA. Details of the different IRB protocols and subsets of these specimens are provided below.

In our prospective collection of ascites samples from these eligible patients (IRB #15066), patients who provided informed consent had their demographic and clinical information collected and de-identified, and ascites were collected before they underwent either diagnostic and/or therapeutic paracentesis in the clinic or diagnostic laparoscopy in the operating room. Under this IRB protocol, RNA sequencing on ascites specimens was performed on two patients with initially diagnosed HGSOC and four patients with recurrent HGSOC. Analysis of IGF2 protein levels was also performed under this IRB protocol using ascites from 10 patients with initial HGSOC and 14 patients with recurrent HGSOC. Some of the patients included in the RNA sequencing analysis were also part of the larger set of patients included in the IGF2 protein analysis.

We also evaluated 23 banked serum specimens collected prior to treatment from a different set of patients with HGSOC who provided informed consent (IRB #3260). Demographic and clinical information for the patients who provided informed consent for donation of their serum were obtained by retrospective review of electronic medical records (IRB #7328).

This study was conducted in accordance with the Scientific Requirements and Research Protocols, Research Ethics Committee, Privacy and Confidentiality, Free and Informed Consent, and Research Registration, Publication, and Dissemination of Results principles of the World Medical Association’s Declaration of Helsinki. Clinical trial number: Not applicable.

### 2.2. RNA Isolation

RNA was isolated from the tumor and ascites using the Zymo Quick RNA extraction kit (Zymo Research, Irvine, CA, USA). Tissues were homogenized using NAVY bead lysis tubes (Next Advanced, Troy, NY, USA; Cat. #NAVYE1-RNA) in RNA/DNA shield (Zymo Research, Irvine, CA, USA). RNA quality and concentration were determined using the Eukaryote Total RNA Pico Series II assay on a Bioanalyzer 2100 (Agilent Technologies, Santa Clara, CA, USA).

### 2.3. RNA Sequencing

Samples with an RNA integrity number (RIN) greater than 6 were sequenced. Stranded RNA sequencing libraries were constructed from 2ng RNA using the poly(A) mRNA isolation kit (NEBNext, Ipswich, MA, USA) and XGen Broad Range RNA Library Prep Kit (Integrated DNA Technologies, Coralville, IA, USA). Each of the libraries was indexed during library construction to allow for multiplex sequencing. Libraries were quantified using a Qubit 4 fluorometer (Invitrogen, Waltham, MA, USA) and checked for size and quality using a 2100 Bioanalyzer (Agilent, Santa Clara, CA, USA). Samples were normalized and pooled onto a 150 paired-end run on a NextSeq 2000 Platform (Illumina, San Diego, CA, USA) to obtain 40–50 M reads per sample. The data were aligned to the Human GRCh38 genome using the DRAGEN RNA v4.2.4 and differential expression v4.2.4 applications in BaseSpace (Illumina, San Diego, CA, USA). The output files included differential expression reports (genes.res), raw gene counts, heat maps, PCA plots, dispersion plots, and MA plots generated by DESeq2, which was incorporated into the DRAGEN differential expression application, at the gene level for each comparison. The data used to produce the heat map were the Log_2_-Normalized expression counts for each sample; thus, the scale was unbounded. Ingenuity Pathway Analysis (IPA, Qiagen, Hilden, Germany) [27] was performed to identify pathways associated with the differentially expressed gene patterns and determine whether they were elevated or reduced in the initial compared to the recurrent samples. Fold changes in genes that were significantly differentially expressed (*p* < 0.05) were entered into and z-scores for different pathways associated with the differential gene expression patterns were generated from the IPA program.

### 2.4. Deconvolution Analysis

Raw counts from the RNA sequencing analysis were used in R 4.3.1 and assessed using the package immunedeconv (Package version: 2.1.0). Analysis was performed using the Microenvironment Cell Populations Counter (MCPCounter Package version: 1.2.0) method to compare samples, cell types, or both. The MCPCounter method allows between- and within-sample comparisons and generates a score in arbitrary units, as described [28,29]. The percentages of cell types were predicted using gene expression data and the ESTIMATE algorithm in R.4.3.1., as described [30].

### 2.5. Pro-Detect Antibody Isotyping

The Pro-Detect Rapid Antibody Isotyping Assay strips were dipped into cell-free ascitic fluid (Thermo Fisher Scientific, Waltham, MA, USA; Cat# A38552). Membrane-based strips are embedded with conjugates to form specific soluble complexes with antibodies present in the samples. The complexes migrate along the membrane and are captured by anti-isotype and class-specific antibodies affixed to the membrane. The results are displayed above a printing on the strip, which indicates isotype and subclass (IgG1, IgG2, IgG3, IgG4, IgA, IgM, and kappa or lambda light chains). Additionally, a red internal control band was included on the strips to indicate that the test was functioning properly. The intensity of bands was scored for each associated antibody class and subclass on the strips from 0 to 3, 0 with no band and 3 indicating intense staining of the band.

### 2.6. IgG Analysis

Ascitic fluid was diluted, and standards, blank, and equal volumes of sample were placed in duplicate wells in a 96-well plate of the IgG ELISA kit. (Abcam #ab195215, Cambridge, UK). The optical density (ODs) endpoints of the assay were corrected by subtracting the average OD of the wells containing the blank. Dilution corrections and linear regression were performed to extrapolate IgG concentrations.

### 2.7. Protein Isolation

Proteins were isolated from ascites cells using Tissue Protein Extraction Reagent (T-PER; Fisher Scientific, Hampton, NH, USA) with a protease inhibitor cocktail. Tissues were homogenized using NAVY bead lysis tubes (Next Advanced, Cat. #NAVYE1-RNA) in the cocktail mix. The protein samples were centrifuged at 14,000× *g* for 10 min at 4 °C, and the protein fraction was collected, frozen, and stored at −80 °C until use. Whole protein lysate concentrations were estimated using bicinchoninic acid (BCA) for ascites samples (Thermo Fisher Scientific, Waltham, MA, USA; Pierce, Cat # 23225).

### 2.8. Insulin-like Growth Factor 2 (IGF2) Enzyme-Linked Immunosorbent Assay (ELISA)

Standards, blanks, and equal serum volumes were placed in duplicate in a 96-well plate (Biotechne R&D Systems #DG200, Minneapolis, MN, USA). The optical density (OD) endpoints of the assay were corrected by subtracting the average OD of the wells containing the blank solution. Dilution corrections and linear regression were performed to extrapolate the IGF2 concentration.

### 2.9. Survival Analysis

The IGF2 concentrations were divided into IGF2 low and IGF2 high expression groups based on the median. The logrank (Mantel–Cox) test [31] and Kaplan–Meier (KM) plot [32] were used to analyze the predictive value of IGF2 low versus high expression with progression-free survival (PFS) or overall survival (OS). Prism version 10.5.0 (GraphPad, Boston, MA, USA) was used to perform the survival analysis.

## 3. Results

### 3.1. Identification of IGF2 and Decreased B-Cells in HGSOC Recurrence

To identify factors associated with HGSOC recurrence, RNA was isolated from ascites specimens of two patients with initial and four with platinum-containing chemotherapy-resistant recurrent HGSOC and sequenced. Chemoresistance was defined as recurrence within six months of platinum-based chemotherapy. The Log_2_-Normalized expression counts for each gene and sample were then compared and sorted by *p*-value. There were 106 genes identified to be differentially expressed between recurrent and initial ascites groups (Appendix A). IGF2 exhibited the highest level of differential expression with greater expression in the recurrent compared to initial ascites groups.

In our analysis, we found that recurrent patients had higher levels of IGF2 in their ascites than initial patients (Figure 1). There was less expression of immunoglobulin variable chain genes in the recurrent samples than in the initial samples. An Ingenuity pathway analysis was conducted on all significant discoveries (*p* > 0.05) and identified multiple immune-regulating pathways (Table 1).

To further evaluate the differential immune gene expression and pathways, a deconvolution analysis of the raw RNA sequencing data was conducted to estimate and compare the percentages of cell types within the samples (Figure 2). The method employs a transcriptome-based computational technique to accurately quantify both immune and non-immune stromal cell populations in a heterogeneous sample, as described in Becht et al. [29]. This analysis utilizes known RNA expression signatures of different immune cells and their activation states to estimate the abundance of immune cells present in the specimens sequenced. The scores demonstrate a correlation with the abundance of the corresponding cell populations across the samples; however, they are expressed in arbitrary units that are directly contingent upon the gene expression within our raw RNA dataset. The B cells and other immune cell types we were interested in characterizing are typically present at low frequencies within biological fluids. Consequently, sequencing was conducted at 40–50 million reads per sample, as opposed to the standard 20 million reads per sample, to enhance the accuracy and quantification of immune cell populations through bioinformatic analysis [29]. The deconvolution analysis estimated an immune cell population associated with suppression in the recurrent ascites based on reduced numbers of memory and plasma B cells, monocytes and M2 macrophages and increased numbers of resting CD4^+^ T memory. The patterns of other immune cell populations, however, were not consistent with an immunosuppressive environment. These included increased CD8^+^ T cells, T follicular helper cells, activated natural killer cells, and activated myeloid dendritic cells.

### 3.2. Immunoglobulins Expressed in Initial Compared to Recurrent Ascites

To validate the sequencing results of immunoglobulin expression in initial and recurrent ascites, we characterized the main classes of immunoglobulins. A Pro-Detect Rapid Antibody Isotyping Assay was performed. The expression of antibody type in each sample was rated 0–3, with 0 being no staining and 3 being high staining. When comparing sample types, we found IgG1 and IgG3 to be the most abundant (Table 2).

To quantify IgG1 expression more accurately, ELISA was performed, and IgG levels were compared between initial and recurrent ascites samples. Additional ascites specimens that had been collected after our RNA sequencing analysis were added to the specimens evaluated for IgG1 expression. No significant difference was observed in IgG levels (Figure 3).

### 3.3. IGF2 Expression Analysis

To validate and quantify the observed differential IGF2 expression at the protein level, we compared IGF2 protein levels in the initial and recurrent ascites (Figure 4A). Because IGF2 is a secreted protein that functions at the cell membrane, we used cell-free ascites fluid in this analysis. The comparison of expression levels between initial and recurrent ascites was significant (*p* = 0.03). Although ascites is collected and useful for prognostic testing, not every patient experiences the accumulation of ascites fluid. Therefore, we evaluated serum samples from patients with HGSOC at initial diagnosis to determine whether serum IGF2 levels are predictive of PFS (Figure 4B). Comparisons of the demographic and clinical characteristics of the patients who donated the serum samples were not significantly different between patients with IGF2 levels below (low IGF2) compared to about the median (high IGF2), which supports that IGF2 expression levels are not due to demographic factors (Table 3).

### 3.4. IGF2 Is Useful for Predicting Survival

Serum samples were sorted into IGF2 high and low groups by the natural division of groups observed at the median value (Figure 4B). There were no significant differences in the demographics of the IFG2-low (below the median) and -high (above the median) groups (Table 4). The pattern of survival observed was that patients with higher IGF2 serum levels experienced worse PFS and OS probability; unfortunately, our study did not have the numbers of patients necessary to provide significant values (Figure 5).

## 4. Discussion

Our study aimed to identify and evaluate genes differentially expressed in initial compared to recurrent HGSOC patient ascites samples. Furthermore, this study sought to validate the findings at the protein level to facilitate the translation of these discoveries into clinically relevant prognostic indicators and survival outcomes.

Our sequencing results revealed significantly higher expression of multiple immunoglobulin and immune regulatory genes in ascites from patients with recurrent HGSOC compared to ascites from patients initially diagnosed with HGSOC. Bioinformatics analysis of the pathways associated with gene expression patterns suggested dysregulation of the immune environment in recurrent samples. The reduction of immunoglobulin variable genes in recurrent samples alludes to fewer B cells being present in the recurrent and platinum-resistant settings. Aspects of our deconvolution analysis support the hypothesis of an immunosuppressive environment in recurrent samples based on the estimated decreased numbers of monocytes, memory and plasma B cells and increased numbers of cytotoxic CD4^+^ T cells in the recurrent samples; however, patterns of other immune cell profiles were not consistent with reduced immune activity in recurrent ascites. These results need to be validated with larger sets of specimens. Because the complex interplay between various immune cells and their relative numbers can alter the ascites immune microenvironment between activity and suppression, functional studies of the immune cell populations in ascites are warranted to decipher how differential immune microenvironments in initial compared to recurrent ascites can affect recurrence.

The most differentially expressed gene in our RNA sequencing analysis was IGF2, which we subsequently validated at the protein level in initial and recurrent ascites fluid samples. IGF2 and its cellular functions have been implicated in primary tumor development, metastasis, immune evasion, and resistance to therapies [33,34]. The stemness of cancer cells is primarily driven by IGF2 signaling, which causes the formation of pre-metastatic niches. In its interaction with the immune system, IGF2 inhibits the apoptosis of various immune cells, thereby suppressing T-cell immunity. This suppression leads to T cell exhaustion, fostering an immunosuppressive environment that ultimately facilitates tumor growth [34].

In ovarian cancer, the expression of IGF2 is associated with HGSOC and advanced stages [35]. There is a correlation between high IGF2 expression and worse progression-free survival [36]. We demonstrated that IGF2 is present at higher levels in recurrent ascites than in initial ascites from patients with HGSOC, and high serum IGF2 levels appear to be associated with worse outcomes in patients with more advanced HGSOC.

In addition to its association with poor patient prognosis, IGF2 is involved in cancer drug resistance, particularly concerning the most commonly used chemotherapy agents in ovarian cancer [37]. Results in experimental models are beginning to define the mechanism of how IGF2 is involved in chemoresistance. Experimentally induced cisplatin resistance in ovarian cancer cell lines was associated with elevation of a long noncoding RNA (lncRNA), IGF2 mRNA and protein and the IGF2/mitogen-activated protein kinase/eukaryotic elongation factor 2 kinase signaling pathway, while interference with this axis promoted cisplatin sensitivity in parental cells [38]. Similarly experimentally induced paclitaxel resistance was associated with elevated IGF2 mRNA, while knockdown of IGF2, but not insulin-like growth factor 1 receptor (IGF1-R), restored sensitivity to paclitaxel in ovarian cancer cell lines [39].

Three therapeutic monoclonal antibodies (AXL1717, MEDI-573, and BI836845) that neutralize IGF1/2 have the potential to counteract IGF2 repression of the immune reaction to cancer and its promotion of chemoresistance in cancer. However, to date no clinical trials of these agents have been conducted in gynecological cancers. In phase II clinical trials, AXL1717 was evaluated in patients with non-small cell lung cancer [40]. Compared to the standard treatment with docetaxel, AXL1717 demonstrated no significant difference in PFS or OS and a lower incidence of side effects. MEDI-573 and BI 836,845 (xentuzumab) have primarily been investigated in the context of metastatic breast cancer. In a phase II clinical trial involving patients with advanced and metastatic breast cancer, the incorporation of xentuzumab into a treatment regimen with exemestane and everolimus did not improve PFS across the entire study population. However, in a pre-specified subgroup of patients without visceral metastasis, the addition of xentuzumab had a favorable impact on PFS [41]. This development prompted a phase II trial focusing on this population; however, the incorporation of xentuzumab with exemestane and everolimus did not enhance progression-free survival in the follow-up study [41]. Although therapeutic agents targeting the IGF1-R and small molecule tyrosine kinase inhibitors within the IGF signaling pathway have been investigated in the context of ovarian cancer, IGF-I/II monoclonal antibodies have not yet been explored. These antibodies may represent potential future therapeutic targets.

The novelty of this study lies in the comparison of ascites from patients initially diagnosed with HGSOC to those with recurrent HGSOC. Other studies of ascites specimens collected from patients with HGSOC or other types of ovarian cancer did not differentiate patients based on the initial versus recurrent status of their disease. Consistent with our results, one of these other studies identified the association of high IGF2 above the median in serum and ascites to be associated with worse prognosis [42]. This other study also differed from ours in that it specifically looked for IGF axis proteins, while our study identified IGF2 from an exploratory analysis of differential gene expression before validating the differential IGF2 protein expression. Another study using machine learning methods to construct a glycolysis-related prognostic model based on publicly available single-cell RNA sequencing data from solid ovarian tumors also identified IGF2 as being prognostically significant [43].

The limitations of this study include the small number of ascites samples in the initial vs. recurrent comparison groups and the serum samples used to evaluate the prognostic significance of IGF2. However, our observation of higher IGF2 gene expression in recurrent ascites specimens than in initial ascites specimens was validated to also occur at the protein level, and our observation of reduced immunoglobulin genes in the recurrent ascites specimens compared to the initial ascites specimens was associated with reduced memory and plasma B-cells, but not naïve B cells, in the recurrent specimens compared to the initial specimens.

## 5. Conclusions

In conclusion, our study identified multiple factors differentially expressed in initial ascites compared to those in recurrent ascites collected from a limited number of patients with HGSOC. Recurrent ascites were associated with significantly higher IGF2 mRNA and extracellular protein levels and reduced immunoglobulin gene expression and activated B cell numbers. These results support further studies of IGF2 inhibition and stimulation of B cell function for the prevention of HGSOC recurrence and the IGF2 prognostic significance using larger groups of patients.

## Figures and Tables

**Figure 1 biomedicines-13-02074-f001:**
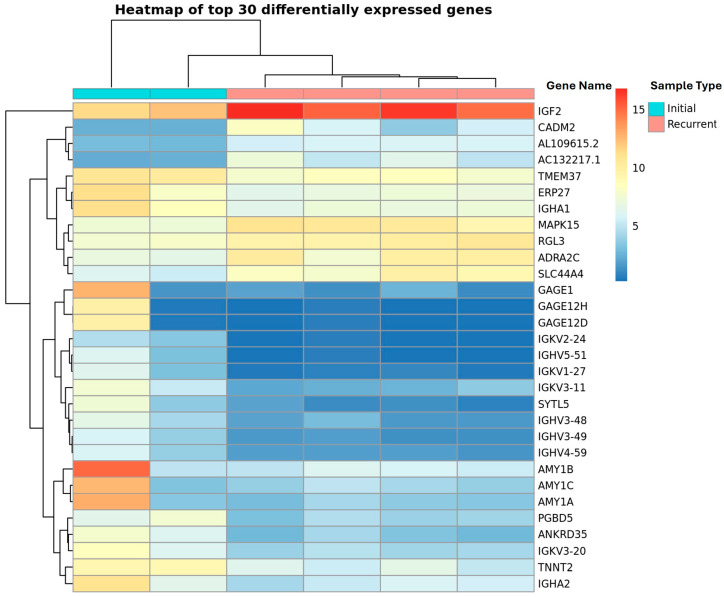
Dendrogram of the top 30 differentially expressed genes in RNA sequencing of initial (N = 2) (teal) versus platinum-resistant recurrent (N = 4) (pink) HGSOC ascites samples. Hierarchical clustering of samples used, and genes found to be differentially expressed are displayed in dendrograms at the top and to the left of the heat map, respectively.

**Figure 2 biomedicines-13-02074-f002:**
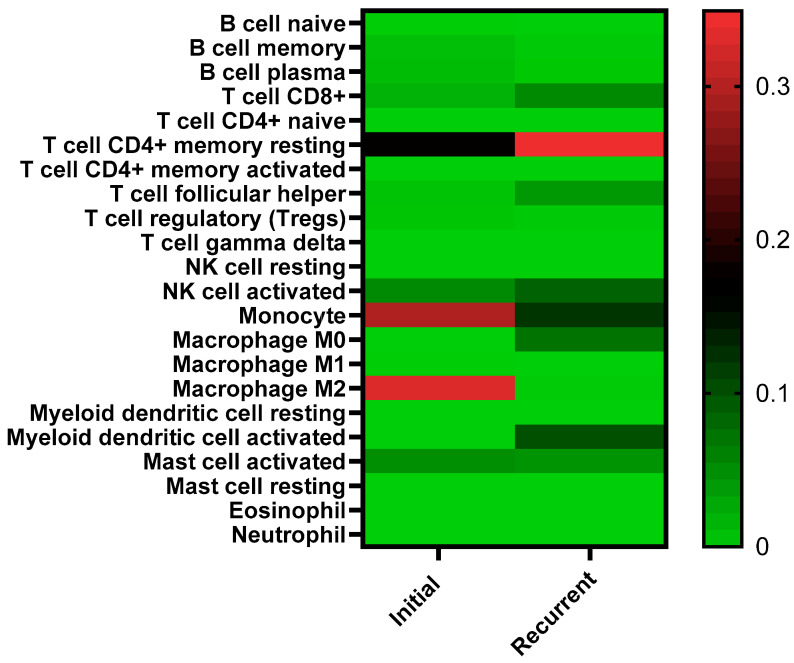
Deconvolution analysis of RNA sequence data to estimate representations of different immune cell populations within the ascites samples. Different cell populations in the initial (N = 2) and recurrent (N = 4) ascites environment are listed to the left and inter-sample abundance is shown in color on a scale of bright green to bright red. The units are arbitrary, where red depicts increased cell type within the sample set and green depicts less cell type in the sample.

**Figure 3 biomedicines-13-02074-f003:**
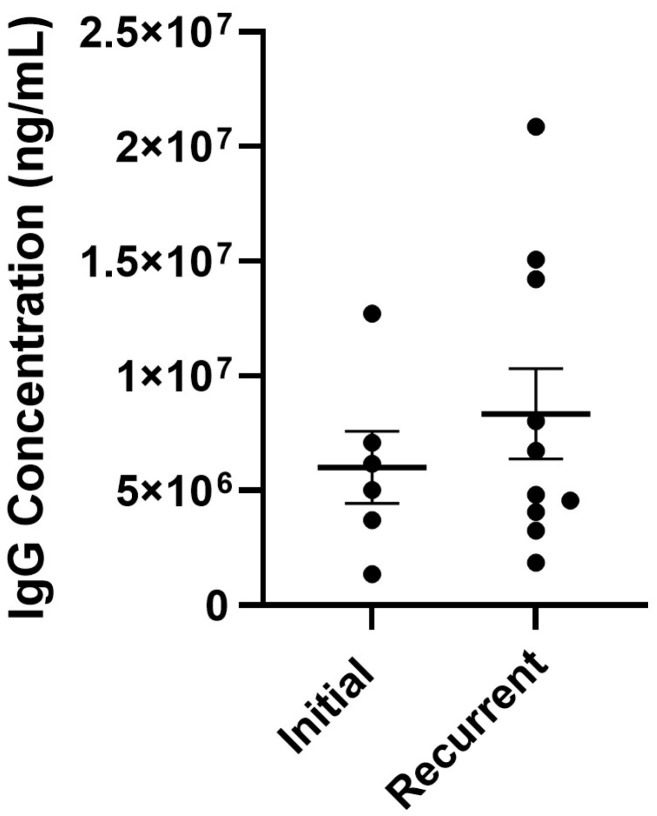
Comparison of IgG1 levels. IgG protein concentration levels in initial (N = 6) and recurrent (N = 10) HGSOC ascites. Each point represents a different ascites sample and its IgG concentration. Samples were normally distributed by Kolmogorov–Smirnov test, unpaired *t* test, *p* = 0.6354.

**Figure 4 biomedicines-13-02074-f004:**
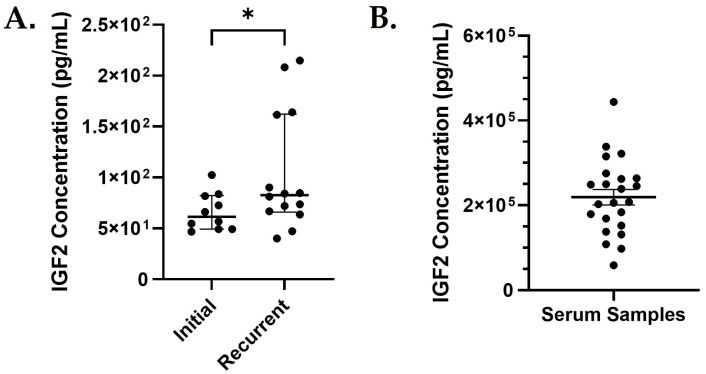
IGF2 Level in Patient Ascites and Serum. (**A**) Comparison of IGF2 concentration in initial (N = 10) vs. recurrent (N = 14) ascites samples. Each point represents a different ascites sample and its corresponding IGF2 concentration. Compared using an unpaired *t* test; * *p* = 0.03. (**B**) IGF2 concentrations measured in serum samples from an independent set of patients diagnosed with HGSOC (N = 23). Each point represents a different serum sample and its IGF2 concentration.

**Figure 5 biomedicines-13-02074-f005:**
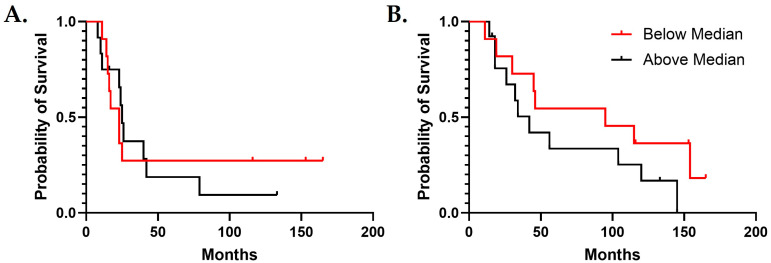
Kaplan–Meier Survival Curves of Patients with HGSOC and Serum IGF2 Expression above versus below the median. (**A**) Comparison of PFS (*p* = 0.87). (**B**) Comparison of OS (*p* = 0.18). The time in months is on the *x*-axis, and the probability of survival is on the *y*-axis.

**Table 1 biomedicines-13-02074-t001:** Ingenuity Pathway Analysis (IPA) of sequencing initial and platinum-resistant recurrent HGSOC patient ascites. Genes with a Q < 0.05 were included in the IPA. The pathway significance cutoff was set at *p* < 0.05, and the Z-score cutoff was >1 or <−1 for significance.

Pathway	z-Score	Gene Names	*p* Value
Communication between Innate and Adaptive Immunity	−4.264	*IGHV3-74*, *IGHV4-4*, *IGHV4-39*, *IGHV4-59*, *IGHV4-61*, *IGHV5-51*, *IGKV1-9*, *IGKV1-27*, *IGKV2-24*, *IGKV3-11*, *IGKV3-20*, *IGLV2-8*, *TLR9*	1.53 × 10^−12^
Binding and Uptake of ligands by Scavenger Receptors	−3	*IGHA1*, *IGHA2*, *IGHV1-69*, *IGHV3-48*, *IGHV4-39*, *IGHV4-59*, *IGKV3-11*, *IGKV3-20*, *IGLV2-8*	2.88 × 10^−10^
Fc gamma receptor (FCGR) dependent phagocytosis	−3	*IGHG2*, *IGHG4*, *IGHV1-69*, *IGHV3-48*, *IGHV4-39*, *IGHV4-59*, *IGKV3-11*, *IGKV3-20*, *IGLV2-8*	5.81 × 10^−9^
Cell Surface interactions at the vascular wall	−3	*IGHA1*, *IGHA2*, *IGHV1-69*, *IGHV3-48*, *IGHV4-39*, *IGHV4-59*, *IGKV3-11*, *IGKV3-20*, *IGLV2-8*	7.54 × 10^−8^
Signaling by the B Cell Receptor	−2.646	*IGHV1-69*, *IGHV3-48*, *IGHV4-39*, *IGHV4-59*, *IGKV3-11*, *IGKV3-20*, *IGLV2-8*	2.88 × 10^−6^
Immunoregulatory interactions between Lymphoid and non-Lymphoid cells	−2.646	*IGHV1-69*, *IGHV3-48*, *IGHV4-39*, *IGHV4-59*, *IGKV3-11*, *IGKV3-20*, *IGLV2-8*	8.95 × 10^−6^
Fc epsilon receptor (FCER) signaling Pathway	−2.464	*IGHV1-69*, *IGHV3-48*, *IGHV4-39*, *IGHV4-59*, *IGKV3-11*, *IGKV3-20*, *IGLV2-8*	1.02 × 10^−5^
Neutrophil Extracellular Trap Signaling Pathway	−1.342	*IGHV1-69*, *IGHV3-48*, *IGHV4-39*, *IGHV4-59*, *IGKV3-11*, *IGKV3-20*, *IGLV2-8*	3.34 × 10^3^
CLEAR Signaling Pathway	−1	*ATP6V0D2*, *BMPR1B*, *MAPK15*, *TLR9*	2.01 × 10^−2^
Neuroinflammation Signaling Pathway	1	*IL34*, *MAPK15*, *PLA2G2D*, *TLR9*	2.83 × 10^−2^
Pathogen-Induced Cytokine Storm Signaling Pathway	1	*CXCL17*, *MAPK15*, *SLC2A1*, *TLR9*	4.61 × 10^−2^

IPA of the significantly differentially expressed genes between the initial and recurrent cancers revealed that multiple immune pathways were significantly downregulated in recurrent ascites compared to those in initial ascites. The most significant pathways also had significant z-scores, representing the fold change and *p*-values of genes within the pathway. Within the significant pathways, most genes were immunoglobulin variable-encoding genes or immune-regulating genes (Table 1).

**Table 2 biomedicines-13-02074-t002:** Pro-Detect Rapid Antibody Isotyping Assay Results. Immunoglobulin class left, 1–5 ascites samples in initial and recurrent HGSOC. Antibody staining classification: 0, no staining; 1, low staining; 2, medium staining; 3, high staining.

		Initial HGSOC Ascites Samples					Recurrent Platinum-Resistant HGSOC Ascites Samples				
	Sample	1	2	3	4	5	1	2	3	4	5
Isotype											
IgG4		0	0	0	0	0	0	0	0	0	0
IgG3		0	0	1	1	2	1	2	0	2	0
IgG2		0	0	1	0	1	1	1	0	2	0
IgG1		3	3	3	0	3	3	3	3	3	3
IgA		1	0	0	0	1	0	1	0	0	0
IgM		0	0	0	0	0	0	0	0	0	0

**Table 3 biomedicines-13-02074-t003:** Demographic Data of Patients with HGSOC who Donated Ascites.

	Initial HGSOC Ascites (N = 10)	Recurrent HGSOC Ascites (N = 14)	* *p*-Value
**Age, N (%)**	<65 years-old: 6 (60)	<65 years-old: 3 (21)	0.09
	≥65 years-old: 4 (40)	≥65 years-old: 11 (79)	
**Initial diagnosis, N (%)**	10 (100)		N/A
**Recurrent diagnosis**		14 (100)	
**Platinum sensitive**		3 patients	
**Platinum resistant**		11 patients	
**Stage at Initial diagnosis, N (%)**	I: 0	I: 0	n.s.
	II: 0	II: 0	
	III: 7 (70)	III: 9 (64)	
	IV: 3 (30)	IV: 5 (36)	
**BMI (kg/m^2^), N (%)**	<18.5: 0	<18.5: 1 (7)	n.s.
	18.5–24.9: 3 (30)	18.5–24.9: 8 (53)	
	25.0–29.9: 2 (20)	25.0–29.9: 3 (21)	
	>30.0: 5 (50)	>30.0: 2 (14)	
**Race, N (%)**	AI/AN: 1 (10)	AI/AN: 1 (7)	n.s.
	White: 9 (90)	White: 13 (93)	

N = number of patients. AI/AN: American Indian/Alaskan Native; n.s.: not significant, *: Fisher’s exact test, N/A: Not Applicable.

**Table 4 biomedicines-13-02074-t004:** Demographic Data of Patients with HGSOC who Donated Serum Categorized as Low vs. High IGF2 Groups.

	IGF2 Serum Concentration High (N = 12)	IGF2 Serum Concentration Low (N = 11)	* *p*-Value
**Age, N (%)**	<65 years-old: 6 (50)	<65 years-old: 7 (64)	n.s.
	≥65 years-old: 5 (42)	≥65 years-old: 4 (36)	
	Unknown: 1 (8)		
**Stage at Initial diagnosis, N (%)**	I: 0	I: 0	n.s.
	II: 0	II: 1 (9)	
	III: 6 (50)	III: 4 (36)	
	IV: 1 (8)	IV: 3 (27)	
	Unknown: 5 (42)	Unknown: 3 (27)	
**BMI (kg/m^2^), N (%)**	<18.5: 1 (8)	<18.5: 0	n.s.
	18.5–24.9: 1 (8)	18.5–24.9: 3 (27)	
	25.0–29.9: 3 (25)	25.0–29.9: 3 (27)	
	>30.0: 1 (8)	>30.0: 1 (9)	
	Unknown: 6 (50)	Unknown: 4 (36)	
**Race, N (%)**	AI/AN: 1 (8)	AI/AN: 0	n.s.
	Black: 2 (16)	Black: 1 (9)	
	White: 9 (75)	White: 10 (91)	

N = number of patients. AI/AN = American Indian/Alaskan Native; n.s. = not significant; * Fisher’s exact test.

## Data Availability

The datasets used and/or analyzed during the current study are available from the corresponding author on reasonable request.

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
