# Peer review of "Increased IGF2 and Immunosuppressive Cell Populations in Ascites of Patients with Recurrent High-Grade Serous Ovarian Cancer"

_biomedicines, 2025, doi:10.3390/biomedicines13092074_

Round 1
Reviewer 1 Report
Comments and Suggestions for Authors
The authors of the manuscript entitled: “Differential Insulin-like Growth Factor 2 and B Cell Activation 2 in Ascites from Patients with Initial compared to Recurrent 3 Ovarian Cancer” display novel prognostic biomarkers for ovarian cancer. In order to prevent recurrence of ovarian cancer, the aim of the sudy was to identify new ascites-linked druggable targets. The experiments focused on ascitic fluid, which act like a liquid tumor microenvironment to facilitate the resistance to treatment.
The manuscript is structured on five parts: a very good introduction, consisting in a synthetic presentation of ovarian cancer- especially HGSOC- recurrence and the role of ascitic fluid in this process; the display of the methods to investigate the differences between the molecular pattern of the cells in initial vs post-treatment ascites; the results alongside an inclusive statistical analysis, the discussion section and the conclusions of the paper.
The paper requires some major revisions:
Methods: Patient Eligibility and Consent- how many patients were recruited in this study? Since details about IRB were not inserted here, a statement regarding the existence of these data in the Institutional Review Board Statement from page 12 could be useful for the readers. The Protein Isolation, RNA Isolation and Deconvultion Analysis subchapters are very brief and some abbreviations were not explained in the text (only on the abbreviation list). It is important to provide details about the way the cell types were identified within the tumor, and in the Figure 2 caption please refer to these data.
Results: in the “Identification of IGF2 and decreased B-Cells in Ovarian Cancer Recurrence” authors should insert a couple phrases about the steps of IGF2 identification as main component.
A broader explanation should be displayed on the different B cell subtypes identification and proportion among other phenotypes; the evidence of their “activation” was not demonstrated or discussed.
I suggest to consolidate with more literature data the Discussion section.
The Conclusions chapter have to contain also the limitations of the study, that of analyzing only 6 cases (as resulted from page 4). Therefore the study is a good start, but further experimental activity is needed in order to establish a new biomarker, such IGF2.
Author Response
Comment 1: Methods: Patient Eligibility and Consent- how many patients were recruited in this study?
Response 1: The details of how many patients were included in each IRB protocol utilized are now listed in the methods section 2.1.
Comment 2: Since details about IRB were not inserted here, a statement regarding the existence of these data in the Institutional Review Board Statement from page 12 could be useful for the readers.
Response 2: We have included details of the patient numbers and types of data in the section on Patient Eligibility and Consent along with the more details about the IRB protocols.
Comment 3: The Protein Isolation, RNA Isolation and Deconvultion Analysis subchapters are very brief and some abbreviations were not explained in the text (only on the abbreviation list).
Response 3: We have carefully reviewed the manuscript and made edits to assure that all abbreviations are explained in the text.
Comment 4: It is important to provide details about the way the cell types were identified within the tumor, and in the Figure 2 caption please refer to these data.
Response 4: We have added more detail to the text and the Figure 2 caption to explain the method used to identify the various cell types in the ascites specimens. (lines 224-233, 239-242)
Comment 5: Results: in the “Identification of IGF2 and decreased B-Cells in Ovarian Cancer Recurrence” authors should insert a couple phrases about the steps of IGF2 identification as main component.
Response 5: We have added more details to explain how IGF2 was identified as the main component. (lines:198-199)
Reviewer 2 Report
Comments and Suggestions for Authors
Article associated with the diagnosis of initial and recurrent ovarian cancer/
The comments on the article are as follows.
- It is necessary to reflect in more detail in the article the used tools of Survival Analysis.
- Table headings must be placed before the tables.
- In Figure 1, it is necessary to add explanations to the images on the left and above, which resemble dendrograms.
- In Figures 3 and 4, it is necessary to add explanations for all the figures used in them – circles, stars, etc. What is shown on the horizontal axis in Figure 4B?
- What are the units of measurement for the data in all tables? What are the numbers in tables 3 and 4?
- In Figure 5, the measured value is incorrectly indicated on the vertical axis: the probability can take values from 0 to 1, it cannot be equal to 100.
- In table S1, the “comparison” column needs to be made wider so that the numbers fit in it without wrapping to the next line.
- Are there any publications on the topic of the article that relate to 2025?
- What is the novelty of the study? What are the limitations of the applicability of the proposed approach?
Comments on the Quality of English Language
The English could be improved to more clearly express the research.
Author Response
Comment 1: It is necessary to reflect in more detail in the article the used tools of Survival Analysis.
Response 1: Details of the specific tools used for the survival analysis have been added to the materials and methods section (section 2.9, Lines 187-192)
Comment 2: Table headings must be placed before the tables.
Response 2: The headings for each table have been placed before the tables throughout the manuscript. (lines 212-214, 249-251, 279, and 294-295)
Comment 3: In Figure 1, it is necessary to add explanations to the images on the left and above, which resemble dendrograms.
Response 3: We have added an explanation of the dendrograms which represent hierarchical clustering in figure 1 legend to help the reader better understand figure 1. (line 209-211)
Comment 4: In Figures 3 and 4, it is necessary to add explanations for all the figures used in them – circles, stars, etc. What is shown on the horizontal axis in Figure 4B?
Response 4: Explanations for both figures 3 and 4 have been added to the figure legends for clarity. The horizontal axis of figure 4B are the serum samples from patients diagnosed with HGSOC that we used to characterize IGF2 expression by ELISA. (lines 258 and 274-277)
Comment 5: What are the units of measurement for the data in all tables? What are the numbers in tables 3 and 4?
Response 5: The numbers in tables 3 and 4 are the number of patients with the associated characteristic in the designated column or row with the percentages in parentheses. We have changed the table headings for each column and row and the footnotes to better convey the information to the reader.
Comment 6: In Figure 5, the measured value is incorrectly indicated on the vertical axis: the probability can take values from 0 to 1, it cannot be equal to 100.
Response 6: The vertical y-axis was changed from percentage to fraction so that the values now fall between 0 and 1.
Comment 7: In table S1, the “comparison” column needs to be made wider so that the numbers fit in it without wrapping to the next line.
Response 7: We have widened the comparison column to ensure the numbers are not wrapped for optimal visual representation. We believe that the final format published for this data will be an Excel file.
Comment 8: Are there any publications on the topic of the article that relate to 2025?
Response 8: We have discussed and cited additional publications related to this topic in the discussion paragraphs addressing the mechanism of IGF2 in ovarian cancer chemoresistance and novelty of this research.
Comment 9: What is the novelty of the study? What are the limitations of the applicability of the proposed approach?
Response 9: We have added text on the novelty and limitations at the end of the discussion section (lines 363-373) .
Reviewer 3 Report
Comments and Suggestions for Authors
Comments to the article “Differential Insulin-like Growth Factor 2 and B Cell Activation in Ascites from Patients with Initial compared to Recurrent Ovarian Cancer” submitted by Laura F Mortan et al. to Biomedicines.
Recurrence in ovarian cancer is routine clinical observation. Depending on the nature of the therapy, the ascites developed in the relapsed cases has appropriate resistance, if any. Ovarian cancers are of different types as per their cell / tissue of origine. Further, the last two decades have stratified ovarian cancer into different molecular classes. Ascites formation is usually observed in high grade serous ovarian adenocarcinoma. Mortan et al have evaluated the ascitic fluids from newly-diagnosed versus recurring ovarian cancer patients. The expression profiling using DESeq2 and immunoassays suggested that IGF2 expression increases in recurring ascites.
Mortan and group have used a clinically important ascites in this study. However, the manuscript is not very well written and planned. The language is not clear and it is difficult to understand what the authors are saying. There are grammatical mistakes all along the document. The is a lack of general flow in the manuscript. The figures have been made casually with no consideration to what is being presented and why. The number of cases in the study is not clear. The numbers do not match in the Figures and Tables. It appears that the authors submitted their first draft and have not read this version before uploading. The authors should improve the manuscript drastically before being considered further.
Comments to the authors.
- Title is not clear. Authors should clearly state the crucial findings here.
- Abstract: It needs to be revised. What type of ovarian cancer is targeted and why is not clear. Number of clinical cases should be mentioned. Which immunoassays? Line 24-26 – what is being attempted here? What is meant by ‘higher’ – the relative percentages will help. Conclusions – inhibition studies are missing in the study. Hence, this sentence should be removed as such work was not performed.
- The authors should judiciously use the template provided by Biomedicines. Material and Methods & Results lack the sub-section numbering.
- Generic sentences should be avoided and appropriate references should be cited for every sentence.
- Line 54. There are different schools of thoughts on the origin of HGSOC. These should either be avoided altogether or all should be mentioned here.
- Authors have casually use of terms - ovarian cancer to HGSOC interchangeably as if they are the same. This is a gross negligence that can be avoided. Ascites is usually in HGSOC and should be mentioned accordingly.
- Lines 75-77. The authors mention that their aim was to study ascites from HGSOC cases, but in methods section, they mention use of retrospective cases where all types of cancers are considered. This creates a confusion what the authors desired and what they actually performed. Please revise and maintain a uniformity.
- Materials and Methods. Lines – 86-90. Was serum and ascites collected for all the patients? Did all the cases have records for samples, tissues and demographic information is not clearly mentioned. Collection of tumor is not mentioned in this section.
- Materials and Methods. Line 106. Apart from concentration, was protein estimated in the each sample?
- Materials and Methods. Except ‘Deconvolution analysis’, all the subsections of methods present the content as if they have been performed for the first time. However, all of them have been performed earlier by various authors. Mortan et al., should refer to them and cite the original author who has worked and established the methods/ protocol for the same. The modification to the original method should be explicitly provided. Self-citation should be avoided.
- Line 134, 139 mentions use of R 4.3.1 while in Line 168 - Survival analysis uses R v4.0.3 which is 2020 release R version. Why did the authors use two different R versions for two assays?
- RNA seq is performed on only 2 patients with no-therapy and 4 patients who had chemo-therapy. This is a very small sample size to comment upon. Further, the internal normalization within the case (use of healthy tissue from the same patient) is missing altogether. Case-control for before-after chemotherapy is missing as well. This was possible in the 2 patients where the treatment was not started.
- Patient data classification should include the nature of the ovarian cancer.
- Figure 1. Typically, DEGs are presented on the scale of 0 and 1 to present significance of the difference. Why have the authors use a current arbitrary scale of 5 to 15?
- Lines 191-193. Description of Ingenuity pathway analysis is too generic. No new information is added here.
- Figure 2. Scale does not present any red color while the heatmap has shades of red coloration for 3 cells. The scale starts from 0 but ends arbitrarily.
- Line 209. What is the semi-quantitative microfluidic assay? There is no mention of this assay in the Material and Methods section. What are the controls? Here, the number of samples for initial and recurrent HGSOC is 5 while the lines 173 mention of only two and four samples.
- Figure 3. The number of HGSOC samples in Initial and Recurrent is 6 and 10, while in line 173 is 2 and 4, while Table 2 has a total of 10 samples.
- Figure 4.A. Number of samples Initial and recurrent is 10 and 14 respectively. What is the unit for y-axis? What is the number of samples for B?
- Table 3. Here, 10 cases of New and 14 cases of Recurrent are listed. What is significance of having races (Asian and Black) when the values are 0 in both the columns?
- Table 4. There is mismatch in figures for BMI cases and Race vis-à-vis Table 3.
- Considering the limited number of cases (which is not clearly evident across the manuscript), the it too early to make statements as in Line 255 to 266.
- Lines 297 to 309. What is the correlation here with the current study?
- Where are the results of the retrospective cases?
- Recent cancer statistics (doi: 10.3322/caac.21871) should have been used instead of current one.
- Format for references is not uniform. Template of the Biomedicine should have been used for writing the references properly. Reports, new articles/ general articles are not written properly and have formatting issues. Some serials that should be checked are 3, 11, 20, 22,23, 28, and 33.
Comments on the Quality of English Language
The authors have written the manuscript very casually. Title and Abstract are not clear at all. The entire manuscript needs to be revised drastically.
Author Response
Comment 1: Title is not clear. Authors should clearly state the crucial findings here.
Response 1: We have changed the title to: Increased IGF2 and Immunosuppressive Cell Populations in Ascites of Patients with Recurrent High Grade Serous Ovarian Cancer
Comment 2: Abstract: It needs to be revised. What type of ovarian cancer is targeted and why is not clear. Number of clinical cases should be mentioned. Which immunoassays? Line 24-26 – what is being attempted here? What is meant by ‘higher’ – the relative percentages will help. Conclusions – inhibition studies are missing in the study. Hence, this sentence should be removed as such work was not performed.
Response 2: The abstract has been modified as specified in the guidance provided by this reviewer.
Comment 3: The authors should judiciously use the template provided by Biomedicines. Material and Methods & Results lack the sub-section numbering.
Response 3: We have consulted the template provided by Biomedicines and have numbered the sub-sections in the materials and methods and the results for better organization and ease of reading.
Comment 4: Generic sentences should be avoided and appropriate references should be cited for every sentence.
Response 4: We have reviewed the manuscript line-by-line and provided more clarity to generic sentences and added relevant citations.
Comment 5: Line 54. There are different schools of thoughts on the origin of HGSOC. These should either be avoided altogether or all should be mentioned here.
Response 5: We have added more details and citations to our discussion on the origins of HGSOC.
Comment 6: Authors have casually use of terms - ovarian cancer to HGSOC interchangeably as if they are the same. This is a gross negligence that can be avoided. Ascites is usually in HGSOC and should be mentioned accordingly.
Response 6: We have carefully reviewed the wording used in this manuscript and have adjusted sentences to provide more clarity on the terms ovarian cancer and HGSOC depending on the context of the sentences.
Comment 7: Lines 75-77. The authors mention that their aim was to study ascites from HGSOC cases, but in methods section, they mention use of retrospective cases where all types of cancers are considered. This creates a confusion what the authors desired and what they actually performed. Please revise and maintain a uniformity.
Response 7: We have clarified in the materials and methods section that only HGSOC cases were included in the analysis.
Comment 8: Materials and Methods. Lines – 86-90. Was serum and ascites collected for all the patients? Did all the cases have records for samples, tissues and demographic information is not clearly mentioned. Collection of tumor is not mentioned in this section.
Response 8: The collection of ascites and serum are two different IRB protocols. The ascites were collected from one cohort of patients and the serum was collected from another cohort of patients at their initial diagnosis and then banked for later use. Each protocol had their own data pull of de-identified demographic data which we used in this manuscript.
We apologize for the mistake and clarify that no solid tumors were used in this study.
Comment 9: Materials and Methods. Line 106. Apart from concentration, was protein estimated in the each sample?
Response 9: Yes, whole protein lysates were used to estimate total protein concentration by BCA. (Line176-177)
Comment 10: Materials and Methods. Except ‘Deconvolution analysis’, all the subsections of methods present the content as if they have been performed for the first time. However, all of them have been performed earlier by various authors. Mortan et al., should refer to them and cite the original author who has worked and established the methods/ protocol for the same. The modification to the original method should be explicitly provided. Self-citation should be avoided.
Response 10: Citations for the original methods for which we did not use commercial kits are now provided.
Comment 11: Line 134, 139 mentions use of R 4.3.1 while in Line 168 - Survival analysis uses R v4.0.3 which is 2020 release R version. Why did the authors use two different R versions for two assays?
Response 11: The package immunedeconv., which uses MCP Counter, and ESTIMATE algorithm works for only R.4.3.1. The description of using R v4.0.3 for the survival analysis was a mistake and we now clarify that GraphPad Software was used instead. (lines:191-192)
Comment 12: RNA seq is performed on only 2 patients with no-therapy and 4 patients who had chemo-therapy. This is a very small sample size to comment upon. Further, the internal normalization within the case (use of healthy tissue from the same patient) is missing altogether. Case-control for before-after chemotherapy is missing as well. This was possible in the 2 patients where the treatment was not started.
Response 12: We appreciate the importance of these comparisons recommended. The patients with initial ovarian cancer who donated specimens did not have ascites present post-chemotherapy and they have not yet recurred. Therefore, we cannot compare before and after chemotherapy. Accumulation of ascites occurs only in the presence of disease. Any peritoneal fluid from patients without cancer will be minimal and unlikely have enough normal epithelial cells to be used for a case-control comparison.
Comment 13: Patient data classification should include the nature of the ovarian cancer.
Response 13: Patient data included the histology (all were HGSOC), grade, stage and treatment and recurrent status of the ovarian cancers. We are not sure what is meant by “nature”.
Comment 14: Figure 1. Typically, DEGs are presented on the scale of 0 and 1 to present significance of the difference. Why have the authors use a current arbitrary scale of 5 to 15?
Response 14: The scale of 0-1 is typically used for non-log-normalized data or fold change with a graph that is using min-max normalization with an odd scale.
The heatmap data is the log2-normalized expression counts for each sample. This means the scaling is not 5-15 but rather the numbers are notating tick marks since the scale is unbounded. We have clarified this in the materials and methods section. (Lines: 134-136)
Comment 15: Lines 191-193. Description of Ingenuity pathway analysis is too generic. No new information is added here.
Response 15: More details were added to the description of the Ingenuity Pathway Analysis. (Lines: 139-141, 216-221)
Comment 16: Figure 2. Scale does not present any red color while the heatmap has shades of red coloration for 3 cells. The scale starts from 0 but ends arbitrarily.
Response 16: This figure was generated using GraphPad Software. The software ends the color legend at the lowest and highest values present in the heatmap. While there are different shades of red due to the transition from black to red based on increasing proportions within samples, there was very little differential red in the heatmap based on the actual values of the data. In lines 224-235 of the results section, we added sentences to provide clarity to this analysis strategy and results presentation.
Comment 17: Line 209. What is the semi-quantitative microfluidic assay? There is no mention of this assay in the Material and Methods section. What are the controls? Here, the number of samples for initial and recurrent HGSOC is 5 while the lines 173 mention of only two and four samples.
Response 17: The term “semi-quantitative microfluidic assay” is a generic term describing the Pro-Detect Rapid Antibody Isotyping Assay. We have removed the term “semi-quantitative microfluidic assay” and replaced it with “Pro-Detect Rapid Antibody Isotyping Assay” to avoid confusion. Additional ascites specimens that had been collected after our RNAseq analysis were added to the specimens evaluated using the Pro-Detect Rapid Antibody Isotyping Assay. (section 2.5, and lines: 245, 249-251)
Comment 18: Figure 3. The number of HGSOC samples in Initial and Recurrent is 6 and 10, while in line 173 is 2 and 4, while Table 2 has a total of 10 samples.
Response 18: The results in Figure 3 were generated using the ascites specimens that were also used in Table 2, as well as additional specimens. We have now clarified this within the text of the Results section. (lines 254-255,258)
Comment 19: Figure 4.A. Number of samples Initial and recurrent is 10 and 14 respectively. What is the unit for y-axis? What is the number of samples for B?
Response 19: The unit for the y-axis has been added to the graph. The number of samples for each figure have been added to the figure legend with N=#. For figure 4B the number of serum samples used is 23. (lines 273-277)
Comment 20: Table 3. Here, 10 cases of New and 14 cases of Recurrent are listed. What is significance of having races (Asian and Black) when the values are 0 in both the columns?
Response 20: The rows referring to Asian and Black patients were deleted from this Table.
Comment 21: Table 4. There is mismatch in figures for BMI cases and Race vis-à-vis Table 3.
Response 21: The patient populations are different in Tables 3 and 4. Table 3 describes the patients who prospectively donated ascites. Table 4 describes patients who donated serum samples to a tissue bank.
Comment 22: Considering the limited number of cases (which is not clearly evident across the manuscript), the it too early to make statements as in Line 255 to 266.
Response 22: This section has been extensively modified to adjust the statements to more accurately reflect the data.
Comment 23: Lines 297 to 309. What is the correlation here with the current study?
Response 23: We have adjusted this paragraph and combined it with a sentence from the previous paragraph to more clearly explain the relevance of the therapeutic antibodies for counteracting IGF2 elevation in HGSOC recurrence.
Comment 24: Where are the results of the retrospective cases?
Response 24: It appears that our description of our prospective ascites collection and our retrospective study of banked serum specimens has made it difficult to understand our manuscript. We have endeavored to provide enhanced description of these studies in the materials and methods and the results section to clarify that the retrospective cases are the serum samples from and independent group of patients with HGSOC.
Comment 25: Recent cancer statistics (doi: 10.3322/caac.21871) should have been used instead of current one.
Response 25: We have removed the 2024 cancer statistics and replaced them with the 2025 statistics, with proper citation.
Comment 26: Format for references is not uniform. Template of the Biomedicine should have been used for writing the references properly. Reports, new articles/ general articles are not written properly and have formatting issues. Some serials that should be checked are 3, 11, 20, 22,23, 28, and 33.
Response 26: We have reformatted the reference section to ensure uniformity. Each of the references were checked in our citing software for formatting and proper display in our manuscript.
Round 2
Reviewer 2 Report
Comments and Suggestions for Authors
The paper can be published in present form.
Author Response
Comment 1: The paper can be published in present form.
Response 1: We appreciate the reviewers consideration.
Reviewer 3 Report
Comments and Suggestions for Authors
The revised manuscript is much better version than its previous version. However, it requires clarifications as described below.
- Line 20. Grammatical mistake. Please revise.
- The total cases for each IRB (#15066, #7328) are not clear.
- The number of cases analyzed by RNA seq is not clear. In abstract, two initial and six recurrent HGSOC samples is mentioned. In Methods, Line 98 mentions two patients with initial and 6 patients with recurrent HGSOC. In Results, Line 196 mentions two patients with initial and four with platinum-containing chemotherapy. Please clarify if all the methods were performed for all clinical samples.
- As per line 100, ascites from 10 patients with initial HGSOC and 14 patients with recurrent HGSOC were available. Why only 5 cases of each are presented in Table 2?
- Format for references 11, 18, 21, 23, 25, 26 is not as per appropriate. At places, the names of the journals are abbreviated / shortened, while at select references, the full name is given. Some references have doi mentioned, while majority don’t have it.
Comments on the Quality of English Language
Grammatical mistakes are present. Please check throughout the manuscript.
Author Response
Comment 1: Line 20. Grammatical mistake. Please revise.
Response 1: The sentence was changed to read “Ascites specimens were collected from patients with HGSOC who provided informed consent.”
Comment 2: The total cases for each IRB (#15066, #7328) are not clear.
Response 2: The numbers have been clarified as shown in the edited paragraph below:
In our prospective collection of ascites samples from these eligible patients (IRB #15066), patients who provided informed consent had their demographic and clinical information collected and de-identified, and ascites were collected before they underwent either diagnostic and/or therapeutic paracentesis in the clinic or diagnostic laparoscopy in the operating room. Under this IRB protocol, RNA sequencing on ascites specimens was performed on two patients with initially diagnosed HGSOC and four patients with recurrent HGSOC. Analysis of IGF2 protein levels was also performed under this IRB protocol using ascites from 10 patients with initial HGSOC and 14 patients with recurrent HGSOC. Some of the patients included in the RNA sequencing analysis were also part of the larger set of patients included in the IGF2 protein analysis.
We also evaluated 23 banked serum specimens collected prior to treatment from a different set of patients with HGSOC who provided informed consent (IRB #3260). Demographic and clinical information for the (23) patients who provided informed consent for donation of their serum were obtained by retrospective review of electronic medical records (IRB #7328).
Comment 3: The number of cases analyzed by RNA seq is not clear. In abstract, two initial and six recurrent HGSOC samples is mentioned. In Methods, Line 98 mentions two patients with initial and 6 patients with recurrent HGSOC. In Results, Line 196 mentions two patients with initial and four with platinum-containing chemotherapy. Please clarify if all the methods were performed for all clinical samples.
Response 3: The number of samples analyzed by RNA sequencing is 2 initial and 4 recurrent. The numbers are now consistent throughout the manuscript.
To provide better clarity about the use of different specimen sets being used for the different methods, we have added the following sentence starting at line 93:
Different sets of specimens were used for various methods. Six ascites specimens were used for the RNA sequencing analysis. Additional ascites specimens were added to this set to equal a total of 24 specimens were used for the IGF2 protein ELISA. A 16-specimen subset of these 24 samples were used to measure IgG protein levels, and a 10-specimen subset of these 24 specimens were used for the isotyping analysis. Twenty-three serum specimens were from a different set of patients with HGSOC were used for another IGF2 protein ELISA. Details of the different IRB protocols and subsets of these specimens are provided below.
As per line 100, ascites from 10 patients with initial HGSOC and 14 patients with recurrent HGSOC were available. Why only 5 cases of each are presented in Table 2?
Table 2 details the quantified results of the Pro-Detect Rapid Antibody Isotyping Assay, which effectively classifies the antibody isotypes of ten samples. Therefore, the samples were categorized into two sets: five representing initial samples and five representing recurrent samples.
Comment 4: Format for references 11, 18, 21, 23, 25, 26 is not as per appropriate. At places, the names of the journals are abbreviated / shortened, while at select references, the full name is given. Some references have doi mentioned, while majority don’t have it.
Response 4: The reference style for MDPI was downloaded and the references mentioned have been changed to consistent styles